# Distributed Orbit Determination for Global Navigation Satellite System with Inter-Satellite Link

**DOI:** 10.3390/s19051031

**Published:** 2019-02-28

**Authors:** Yuanlan Wen, Jun Zhu, Youxing Gong, Qian Wang, Xiufeng He

**Affiliations:** 1School of Earth Sciences and Engineering, Hohai University, Nanjing 210098, China; wqaloha@139.com; 2State Key Laboratory of Astronautic Dynamics, Xi’an 710043, China; zhujun9306@126.com; 3Undergraduate School, National University of Defense Technology, Changsha 410073, China; 13874804178@139.com

**Keywords:** inter-satellite link, whole-constellation centralized extended Kalman filter, distributed orbit determination, iterative cascade extended Kalman filter, increased measurement covariance extended Kalman filter, balanced extended Kalman filter

## Abstract

To keep the global navigation satellite system functional during extreme conditions, it is a trend to employ autonomous navigation technology with inter-satellite link. As in the newly built BeiDou system (BDS-3) equipped with Ka-band inter-satellite links, every individual satellite has the ability of communicating and measuring distances among each other. The system also has less dependence on the ground stations and improved navigation performance. Because of the huge amount of measurement data, the centralized data processing algorithm for orbit determination is suggested to be replaced by a distributed one in which each satellite in the constellation is required to finish a partial computation task. In the present paper, the balanced extended Kalman filter algorithm for distributed orbit determination is proposed and compared with the whole-constellation centralized extended Kalman filter, the iterative cascade extended Kalman filter, and the increasing measurement covariance extended Kalman filter. The proposed method demands a lower computation power; however, it yields results with a relatively good accuracy.

## 1. Introduction

For the global navigation satellite system (GNSS), the master control station (MCS) currently collects the satellite to monitor station measurement data, estimates the satellite ephemeris and clock offsets, and generates a time stream of navigation messages. The messages are then uploaded to satellites by ground antennas to broadcast to the user community [1]. However, the MCS as well as the other ground-based segments including monitor stations and ground antennas have the risk of destruction during a warfare or natural disaster. This is the case especially for the monitor stations which are distributed globally for increasing the accuracy of satellite orbit determination [2]. In order to enhance the viability of satellite navigation systems under the potentially fatal conditions, as early as in the 1980s, autonomous navigation techniques using inter-satellite link (ISL) measurements without support from the MCS were investigated for the global positioning system (GPS) [3]. If the ISL measurement was the only source for orbit determination and time synchronization, the datum mark would be insufficient [4]. This problem can be addressed by setting up a few ground anchorage stations (GASs) which provide a reference coordinate system and a time system [5]. Combining both the ISL and satellite-to-GAS measurements, the autonomous navigation system has several features: firstly, data processing will be completed by satellite onboard computers rather than the MCS; secondly, the GASs can be considered as pseudo-satellites; and finally, the globally allocated monitor stations can be replaced by a few domestic GASs [3].

The centralized data processing technique, which is widely applied in the current autonomous navigation constellation, collects the ISL and satellite-to-GAS measurement data, combines all the satellite state vectors (including satellite orbit, clock error parameters, etc.) into one matrix, and computes the optimal state vector for each satellite by a central satellite on-board computer. As a result, the associated state vector covariance matrix could be very large and the method would require vast computation power [6]. It is worth mentioning that, in this case, every satellite is observed indirectly by other satellites.

On the other hand, in the distributed data processing algorithm, computation is broken down and assigned to each satellite. Each satellite is required to deal with self-related ISL measurements and local state vectors. In this way, the number of measurement equations and dimension of the state vectors are reduced. As a result, the computational amount of the whole system is decreased considerably. Moreover, the accuracy of the orbit ephemeris and clock offsets calculated by the distributed data processing method could have the same level of accuracy as the results of centralized data processing.

With the measurements from ISLs and satellite-to-GASs, the autonomous navigation constellation constitutes an extremely complex system. Each satellite has to finish the task of ISL measurement, data processing, and communicating. A practical algorithm for autonomous orbit determination is still under development. Based on the methods of the iterative cascade extended Kalman filter (ICEKF) and the increased measurement covariance extended Kalman filter (IMCEKF), a new distributed method, the balanced extended Kalman filter (BEKF), is proposed in this paper. Together with whole-constellation centralized extended Kalman filter (WCCEFK), four different autonomous navigation algorithms are conducted in simulations for comparisons of accuracy and computation loads.

## 2. Overview of Orbit Determination Algorithms

Ananda [7] proposed the framework of an autonomous navigation system without a MCS for the first time and validated the system by simulations. Rajan [3] introduced various autonomous navigation algorithms and presented preliminary on-orbit experiment results.

In the designing of the GPS Block IIR, the ISLs, programmable microprocessors, and redundant management were carried out. The following two major features are critical to ensure high precision autonomous navigation [3,7]:the ISL communications and measurements in ultra high frequency (UHF) band;a high-precision autonomous navigation algorithm which is adapted to the computing capacity of the satellite on-board computers.

A time division multiple access (TDMA) system, which has two frames, was employed for ranging and data transmission. In a ranging frame, each satellite is assigned a 1.5-s slot to make pseudo-range (PR) measurements with the visible satellites. Two frequencies were used in the measurement for ionospheric delay corrections. In a data frame, a 1.5-s slot was appointed to each satellite to transmit data that includes the PR measurements, the estimated satellite state vector, and the associated covariance matrix to all visible satellites.

The GPS Block IIF follows the design of the GPS Block IIR and improved the performance of ISL measurements and on-board data processing. Without contacting with the ground system, Block IIF can operate about 60 days in the autonomous navigation mode and provides navigation messages which are corrected by ISL measurements with a 3-m user range error (URE) (URE is a root mean square value and does not consider the impact of the polar motion and UT1). However, it is difficult to establish a precise prediction system because of the irregular polar motion and uncertain UT1. Therefore, in autonomous navigation mode, the URE is far greater than 3 m after a 60-day duration [8].

For GPS III, each satellite in the constellation will have the ability of ISL measurements and communications. It is designed that once there are enough satellites in the orbit, the GPS constellation will be able to operate autonomously in wartimes, but currently it is still under investigation [9].

Galileo navigation system is also planned to employ an autonomous navigation algorithm based on ISLs [10]. The spatial orientation problem is solved by the combination of the ISL and satellite-to-ground measurements. The simulation for the Galileo system showed that URE is on the level of decimeters [10], which is better than that of GPS.

For the distributed navigation algorithms, several techniques including the iterative cascade extended Kalman filter (ICEKF), increased measurement covariance EKF (IMCEKF), and Schmidt-Kalman Filter (SKF) are discussed by Schmidt, Park, and Ferguson [11,12,13]. The ICEKF is employed to processes a large number of space-borne GPS measurement data and a small quantity of ISL measurement data for low-Earth orbit formation flying satellites. In this method, the computation process will iterate for 3 to 4 times for convergence, and a good orbit accuracy is presented. For the method of IMCEKF, amendments are made based on ICEKF and it presents a better performance. On the other hand, the SKF yields results with less accuracy compared to IMCEKF while processing the GNSS ISL measurement data [4]. Recently, the International Association of Geodesy (IAG) initiated a GPS Dancer project which develops a distributed data processing algorithm to analyze the precision of GPS [14].

In China, distributed orbit determination and time synchronization algorithms based on ISL measurements were studied [15,16,17,18,19,20,21]. Distributed autonomous ephemeris updating was discussed for navigation satellites [22]. There are also many studies on developing higher precision orbit determination methods for new BeiDou (BDS-3) experimental satellites in the literature [23,24,25,26]. Currently, the developments for autonomous orbit determination, time synchronization, and autonomous operation and management are being widely investigated. More progress on designing an efficient distributed data processing algorithm, however, needs to be made.

## 3. Fundamental Equations for Measurement and Motion

### 3.1. Equations for Measurement

The position vectors, ${\vec{r}}^{i}=[{\begin{array}{ccc} x^{i} & y^{i} & z^{i}] \end{array}}^{T}$, velocity vectors, ${\vec{v}}^{i}=\begin{array}{ccc} {}[v_{x}^{i} & v_{y}^{i} & v_{z}^{i} \end{array}{]}^{T}$, and dynamic parameter vector, $x_{D}^{i}$, constitute the state vector which needs to be estimated:(1)$$X_{}^{i}={\left[{\vec{r}}_{}^{i}{}^{T}\;{\vec{v}}_{}^{i}{}^{T}\;x_{D}^{i}{}^{T}\right]}^{T}$$
where the superscript *T* denotes the transpose of a matrix, and the superscript *i* denotes that it is for the *ith* satellite. The reference values for the state vector are stored in $X_{}^{i}{}^{*}$, and the improving values for the state vector are written in:(2)$$\delta x_{}^{i}=X_{}^{i}-X_{}^{i}{}^{*}$$

After correcting the hardware delay, ionospheric delay, relativistic effect, multi-path effect, and the antenna phase center offset [4], two one-way ISL PRs between the *ith* satellite and *jth* satellite need to be translated into the same measurement epoch (e.g., ranging frame epoch *t*). The PR equations are:(3)$$\rho_{}^{i\to j}(t)=c\cdot [\delta t_{}^{j}(t)-\delta t_{}^{i}(t)]+d(X_{}^{j}(t),X_{}^{i}(t))+\varepsilon_{}^{i\to j}(t)$$
(4)$$\rho_{}^{j\to i}(t)=c\cdot [\delta t_{}^{i}(t)-\delta t_{}^{j}(t)]+d(X_{}^{j}(t),X_{}^{i}(t))+\varepsilon_{}^{j\to i}(t)$$
where, *c* is the speed of light, $\delta t_{}^{i}(t)$ and $\delta t_{}^{j}(t)$ are clock errors for the *ith* satellite and the *jth* satellite, respectively, $\varepsilon_{}^{i\to j}(t)$ and $\varepsilon_{}^{j\to i}(t)$ are the measurement errors, and $d(X_{}^{j}(t),X_{}^{i}(t))=$$\sqrt{{\left(x_{}^{i}-x_{}^{j}\right)}^{2}+{\left(y_{}^{i}-y_{}^{j}\right)}^{2}+{\left(z_{}^{i}-z_{}^{j}\right)}^{2}}$ is the geometric distance between the two satellites. Combining Equations (3) and (4), the distance measurement equation that only contains orbit parameters is derived as:(5)$$\rho_{}^{ij}(t)=\left[\rho_{}^{j\to i}(t)+\rho_{}^{i\to j}(t)\right]/2=d(X_{}^{j}(t),X_{}^{j}(t))+\varepsilon_{}^{ij}(t)$$
where, $\varepsilon_{}^{ij}(t_{k})=[\varepsilon_{}^{i\to j}(t_{k})+\varepsilon_{}^{j\to i}(t_{k})]/2$. Subtracting Equation (4) from Equation (3), the time measurement equation that only contains clock error parameters is deduced as:(6)$$\rho_{clock}^{ij}(t)=\left[\rho_{}^{j\to i}(t)-\rho_{}^{i\to j}(t)\right]/2=c\cdot [\delta t_{}^{j}(t)-\delta t_{}^{i}(t)]+\left[\varepsilon_{}^{j\to i}(t)-\varepsilon_{}^{i\to j}(t)\right]/2$$

Following the steps above, the distance measurements and the clock bias measurements are decoupled. The orbit ephemeris and clock offsets can therefore be estimated independently.

Linearizing Equation (5) with Taylor expansion at the reference state vector $X_{}^{i}{}^{*}$ and $X_{}^{j}{}^{*}$ yields:(7)$$\rho_{}^{ij}(t)=d\left(X_{}^{i}{}^{*},X_{}^{j}{}^{*}\right)+{\frac{\rho_{}^{ij}(t)}{\partial X_{}^{i}}|}_{X_{}^{i}{}^{*}}\delta x_{}^{i}+{\frac{\rho_{}^{ij}(t)}{\partial X_{}^{j}}|}_{X_{}^{j}{}^{*}}\delta x_{}^{j}+\cdots +\varepsilon_{}^{ij}(t)$$

After that, Equation (7) is converted into a linear measurement equation:(8)$$z_{}^{ij}(t)=H_{}^{i}\delta x_{}^{i}(t)+H_{}^{j}\delta x_{}^{j}(t)+\varepsilon_{}^{ij}(t)$$
where $z_{}^{ij}(t)=\rho_{}^{ij}(t)-d\left(X_{}^{i}{}^{*},X_{}^{j}{}^{*}\right)$ is the innovation. $H_{}^{i}$ and $H_{}^{j}$ are the measurement matrices.
(9)$${H_{}^{i}=\frac{\rho_{}^{ij}(t)}{\partial X_{}^{i}}|}_{X_{}^{i}{}^{*}}={\left[\begin{array}{ccc} \frac{x_{}^{i}-x_{}^{j}}{d\left(X_{}^{i}{}^{*},X_{}^{j}{}^{*}\right)} & \frac{y_{}^{i}-y_{}^{j}}{d\left(X_{}^{i}{}^{*},X_{}^{j}{}^{*}\right)} & \frac{z_{}^{i}-z_{}^{j}}{d\left(X_{}^{i}{}^{*},X_{}^{j}{}^{*}\right)}\; \end{array}\right]}_{X_{}^{i}{}^{*}}$$
(10)$${H_{}^{j}=\frac{\rho_{}^{ij}(t)}{\partial X_{}^{j}}|}_{X_{}^{j}{}^{*}}=-H_{}^{i}$$

Similarly, GAS can be considered as a pseudo-satellite. The PR needs to be corrected by an extra tropospheric delay compared to a normal satellite. The distance measurement equation that contains orbit parameters between the *gth* GAS and *ith* satellite is derived as:(11)$$\rho_{}^{ig}(t)=\left[\rho_{}^{g\to i}(t)+\rho_{}^{i\to g}(t)\right]/2=d(X_{}^{i}(t),X_{}^{g}(t))+\varepsilon_{}^{ig}(t)$$

In the Equation (11), the array of state parameters of GAS $X_{}^{g}(t)$ is known; only the state vector of the *ith* satellite is unknown. The reference ground coordinate is hence introduced into the satellite state by Equation (11), which overcomes the lack of the datum mark in data processing while only ISL measurements are utilized. The current linearized measurement equation, which is similar to Equation (8), becomes:(12)$$z_{}^{ig}(t_{})=H_{}^{i}\delta x_{}^{i}(t)+\varepsilon_{}^{ig}(t)$$

### 3.2. Equations for Motion

Satellites can be affected by a variety of factors when operated in orbit. For navigation satellites, only the gravitational forces, solar radiation pressure, and relativistic effects are considered [27]. The gravitational forces include the attractions from the earth, the moon, and other planets in the solar system. The dynamic equation for the *ith* satellite can be written as:(13)$${\dot{X}}_{}^{i}(t)=f^{c}(X_{}^{i},w_{}^{i})$$
where $f^{c}$ is a continuous function, $w_{}^{i}$ is the system disturbances that have the following properties:(14)$$\begin{array}{cc} E\left[w_{}^{i}(t)\right]=0, & E\left[w_{}^{i}(t){\left(w_{}^{i}(\tau )\right)}^{T}\right]=\{\begin{array}{cc} Q_{}^{i}(t) & t=\tau \\ 0 & t\neq \tau \end{array} \end{array}$$
E[⋅] denotes the expected value, and $Q_{}^{i}(t)$ is a covariance matrix which is symmetric, non-negative, and definite. Here, the system disturbances are simulated by Gaussian white noise. Because the continuous function $f^{c}$ and the system disturbance $w_{}^{i}$ are not coupled with each other, Equation (13) can be written as:(15)$${\dot{X}}_{}^{i}(t)={ℱ}^{c}(X_{}^{i}(t))+Gw_{}^{i}$$
In which $G={\left[\begin{array}{ccc} 0 & I & 0 \end{array}\right]}^{T}$ is the coefficient matrix, I is the identity matrix, and 0 is the zero matrix. Equation (15) is then linearized by Taylor expansion at the reference state vector $X^{i*}$:(16)$${\dot{X}}_{}^{i}(t)={ℱ}^{c}(X_{}^{i}{}^{*}(t))+{\frac{\partial {ℱ}^{c}(X_{}^{i})}{\partial X_{}^{i}}|}_{X_{}^{i}{}^{*}}(X_{}^{i}(t)-X_{}^{i}{}^{*}(t))+\cdots +Gw_{}^{i}$$

From the equation above, the state increments are then derived as:(17)$$\delta {\dot{x}}_{}^{i}(t)=F(t)\delta x_{}^{i}(t)+Gw_{}^{i}$$
where $\delta \dot{x}=\dot{X}-{ℱ}^{c}(X_{}^{i}{}^{*}(t))$, and F(t) is the dynamic partial derivative matrix [6]:(18)$$F(t)={\frac{\partial {ℱ}^{c}(X)}{\partial X}|}_{X^{*}}={\left[\begin{array}{l} \begin{array}{ccc} 0 & I & 0 \end{array}\\ \begin{array}{ccc} \frac{\partial {ℱ}^{c}}{\partial r} & \frac{\partial {ℱ}^{c}}{\partial \dot{r}} & \frac{\partial {ℱ}^{c}}{\partial p_{D}} \end{array}\\ \begin{array}{ccc} 0 & 0 & 0 \end{array} \end{array}\right]}_{|X^{*}}$$

Equation (17) is the state equation of the stochastic linear continuous systems and its general solution is:(19)$$\delta x_{}^{i}(t)=\Phi_{}^{i}(t,t_{0})\delta x_{0}^{i}(t)+G\int_{t_{0}}^{t}\Phi_{}^{i}(t,\tau )w_{}^{i}(\tau )d\tau$$
in which $\Phi_{}^{i}(t,t_{0})$ is the system state transition matrix and the solution of the following equations:(20)$$\begin{array}{cc} {\dot{\Phi }}_{}^{i}(t,t_{0})=F(t)\Phi_{}^{i}(t,t_{0}), & \Phi_{}^{i}(t_{0},t_{0})=I \end{array}$$
where I is the identity matrix with the same dimensions as dynamics matrix, F(t).

The state transition matrix $\Phi_{}^{i}(t,t_{0})$ has the following features:(21)$$\begin{array}{cc} \Phi_{}^{i}(t,\tau )\Phi_{}^{i}(\tau ,t_{0})=\Phi_{}^{i}(t,t_{0}), & {\left[\Phi_{}^{i}(t,\tau )\right]}^{-1}=\Phi_{}^{i}(\tau ,t) \end{array}$$

In the actual computation process, discretization needs to be implemented for Equation (19).
(22)$$\delta x_{k}^{i}=\Phi_{}^{i}(t_{k},t_{k-1})\delta x_{k-1}^{i}+G\int_{t_{k-1}}^{t_{k}}\Phi_{}^{i}(t_{k},\tau )w_{k-1}^{i}(\tau )d\tau$$

In a sampling interval from $t_{k-1}$ to $t_{k}$, the white noise $w_{k-1}^{i}(\tau )$ can be considered as a constant. The integral coefficient denotes that:(23)$$G_{k}^{i}=G\int_{t_{k-1}}^{t_{k}}\Phi (t_{k},\tau )d\tau$$

For simplicity, the white noise will be denoted as $w_{k-1}^{i}$ in the following. The discretized state equation derived from Equation (22) then is:(24)$$\delta x_{k}^{i}=\Phi_{k-1}^{i}\delta x_{k-1}^{i}+G_{k}^{i}w_{k-1}^{i}$$
where $\Phi_{k-1}^{i}$ denotes the state transition matrix from $t_{k-1}$ to $t_{k}$. According to Equation (24), the predicted state covariance matrix is:(25)$${\bar{P}}_{k}^{i}=\Phi_{k-1}^{i}{\hat{P}}_{k-1}^{i}\Phi_{k-1}^{iT}+G_{k-1}^{i}Q_{k-1}^{i}G_{k-1}^{iT}$$

## 4. Whole-Constellation Centralized Extended Kalman Filter

The whole-constellation centralized extended Kalman filter (WCCEKF) is one of the centralized data processing methods. According to this method, a main satellite and a back-up satellite are assigned to complete the task of data processing. The other satellites in the constellation need to send their measurement data, state vectors, and corresponding covariance matrices to the main satellite for orbit determinations.

For all the satellites in the constellation, the states and corresponding improving values from Equations (1) and (2) are collected and stored in a state vector $X_{k}^{}$ and an improving values vector $\delta x_{k}^{}$ [6,28], as in
(26)$$X_{k}^{}={\left[\begin{array}{cccc} (X_{}^{1}{)}^{T} & (X_{}^{2}{)}^{T} & \cdots \;(X_{}^{i}{)}^{T} & \cdots \;(X_{}^{n}{)}^{T} \end{array}\right]}^{T}$$
(27)$$\delta x_{k}^{}={\left[\begin{array}{cccc} (\delta x_{k}^{1}{)}^{T} & (\delta x_{k}^{2}{)}^{T} & \cdots \;(\delta x_{k}^{i}{)}^{T} & \cdots \;(\delta x_{k}^{n}{)}^{T} \end{array}\right]}^{T}$$
where *n* is the number of satellites in the system. In this way, the state equation for all the satellites can be obtained through Equation (24).
(28)$$\delta x_{k}^{}=\Phi_{k-1}^{}\delta x_{k-1}^{}+G_{k}^{}w_{k-1}^{}$$

The state transition matrix $\Phi_{k}^{}$ and integral coefficient matrix $G_{k}^{}$ are diagonal matrices and can be expressed as:(29)$$\Phi_{k-1}^{}=diagonal\left[\begin{array}{cccc} \left(\Phi_{k-1}^{1}\right) & \left(\Phi_{k-1}^{2}\right) & \cdots \;(\Phi_{k-1}^{i}) & \cdots \;(\Phi_{k-1}^{n}) \end{array}\right]$$
(30)$$G_{k}^{}=diagonal\left[\begin{array}{cccc} \left(G_{k}^{1}\right) & \left(G_{k}^{2}\right) & \cdots \;(G_{k}^{i}) & \cdots \;(G_{k}^{n}{)}^{T} \end{array}\right]$$

The state noise vector, $w_{k-1}^{}$, stores the noise for all the satellites in the constellation,
(31)$$w_{k-1}^{}={\left[\begin{array}{cccc} \left(w_{k-1}^{1}\right) & \left(w_{k-1}^{2}\right) & \cdots \;(w_{k-1}^{i}) & \cdots \;(w_{k-1}^{n}) \end{array}\right]}^{T},$$
and it has the statistical characteristics as follows:(32)$$\begin{array}{cc} E\left[w_{k-1}^{}\right]=0, & E\left[w_{k-1}^{i}{\left(w_{k-1}^{j}\right)}_{}^{T}\right]=\{\begin{array}{cc} Q_{k-1}^{i} & i=j\\ 0 & i\neq j \end{array} \end{array}\;$$
(33)$$E\left[w_{k-1}^{}w_{k-1}^{T}\right]=Q_{k-1}^{}=diagonal\left[\begin{array}{cccc} Q_{k-1}^{1} & Q_{k-1}^{2} & \cdots \;Q_{k-1}^{i} & \cdots \;Q_{k-1}^{n} \end{array}\right]$$

The measurement equation, which is a combination of Equations (8) and (12), is then derived as:(34)$$z_{k}^{}=H_{k}^{}\delta x+\varepsilon_{k}^{}$$
where $H_{k}^{}={\left[0\;\cdots \;{\left(H_{}^{i}\right)}^{T}\cdots \;0\;\cdots \;{\left(H_{}^{j}\right)}^{T}\cdots \;0\right]}^{T}$, $z_{k}^{}=z_{k}^{ij}(t_{k})$, and $\varepsilon_{k}^{}=\varepsilon_{}^{ij}(t_{k})$ for the *ith* satellite and *jth* satellite with ISL; $H_{k}^{}={\left[0\;\cdots \;{\left(H_{}^{i}\right)}^{T}\cdots \;0\right]}^{T}$, $z_{k}^{}=z_{k}^{ig}(t_{k})$, and $\varepsilon_{k}^{}=\varepsilon_{}^{ig}(t_{k})$ for GAS-to-*ith* satellite measurements. Next, the measurement covariance matrix $R_{k}^{}=E(\varepsilon_{k}^{}\varepsilon_{k}^{T})$, the initial state vector ${\bar{X}}_{0}=E(X_{0}^{*})$, and the initial state vector covariance ${\bar{P}}_{0}=E[X_{0}^{*}X_{0}^{*T}]$ are defined. Finally, the method of WCCEKF that combines the satellite-to-satellite measurements and satellite-to-GAS measurements can be expressed as [29]:(35)$${\bar{P}}_{k}^{}=\Phi_{k-1}{\hat{P}}_{k-1}^{}\Phi_{k-1}^{T}+G_{k-1}Q_{k-1}G_{k-1}^{T}$$
(36)$$K_{k}={\bar{P}}_{k}^{}H_{k}^{T}{\left(H_{k}^{T}{\bar{P}}_{k}^{}H_{k}+R_{k}\right)}^{-1}$$
(37)$$\delta {\hat{x}}_{k}^{}=K_{k}z_{k}$$
(38)$${\hat{X}}_{k}^{}={\bar{X}}_{k}^{}+\delta {\hat{x}}_{k}^{}$$
(39)$${\hat{P}}_{k}^{}=\left(I-K_{k}H_{k}\right){\bar{P}}_{k}^{}$$

The dimension of state vector $X_{k}^{}$ is
(40)$$N_{W}^{}=6n+\sum_{i=1}^{n}D_{{}_{}}^{i}$$
where $D_{{}_{}}^{i}$ is the number of dynamic parameters for the *ith* satellite.

In the method of WCCEKF, each satellite is correlated with the other satellites through the state vector covariance matrix which has the dimension of $N_{W}^{}\times N_{W}^{}$. Furthermore, matrix $\left(H_{k}^{T}{\bar{P}}_{k}^{}H_{k}+R_{k}\right)$ with the dimension of m×m (m is the dimension of the measurement vector) needs to be inversed during the process, and a huge computation amount is expected. The computation amount for a process of WCCEKF is:(41)$$4N_{W}^{2}(N_{W}^{2}-1)+(N_{W}^{}-1)N_{W}^{}(N_{W}^{}+1)/6+(2N_{W}^{2}+7N_{W}^{}+1)\times m$$

If the WCCEKF algorithm is employed, the on-board computer of the main satellite would need to process all the ISL measurement and satellite-to-GAS measurement data to finish the task of orbit determination and navigation message generation for satellites in the constellation. Due to the huge computation amount and great complexity of communication, WCCEKF is difficult to be implemented in a satellite constellation with limited on-board computation ability.

In addition, the WCCEKF is also vulnerable. Once the main satellite and its backup satellite failed, the entire navigation constellation would stop working. To avoid the drawbacks in the WCCEKF method, many researches nowadays are focusing on developing a distributed data processing algorithm.

## 5. Distributed Orbit Determination

For distributed orbit determination algorithms based on ISL, data processing is assigned to each satellite. In this process, each satellite collects the ISL measurement data with respect to its visible satellite and estimates the self-related state vectors.

### 5.1. Reduced-Order Iterative Cascade EKF

For *ith* and *jth* satellites with ISL measurements, the iterative cascade EKF (ICEKF) [12] assumes that the state vector $X_{}^{j}$ of the *jth* satellite is known. Thus, the measurement equation, which is similar to Equation (34), is derived as:(42)$$z_{k}^{i}=H_{k}^{i}\delta x_{k}^{i}+\varepsilon_{k}^{i}$$

For ISL measurement, the innovation is $z_{k}^{i}=z_{k}^{ij}(t_{k})$, the measurement error is $\varepsilon_{k}^{i}=H_{k}^{j}\delta x_{k}^{j}+\varepsilon_{}^{ij}(t_{k}^{})$ and measurement covariance matrix is $R_{k}^{i}=E[\varepsilon_{k}^{i}{\left(\varepsilon_{k}^{i}\right)}_{}^{T}]=E[\varepsilon_{}^{ij}(t_{k}^{})\varepsilon_{}^{ij}{\left(t_{k}^{}\right)}_{}^{T}]=R_{k,ISL}^{i}$. For satellite-to-GAS measurement, the innovation is $z_{k}^{i}=z_{k}^{ig}(t_{k})$, the measurement error is $\varepsilon_{k}^{i}=\varepsilon_{}^{ig}(t_{k}^{})$, and measurement covariance matrix is $R_{k}^{i}=E(\varepsilon_{k}^{i}{\left(\varepsilon_{k}^{i}\right)}_{}^{T}]=E(\varepsilon_{}^{ig}(t_{k}^{})\varepsilon_{}^{ig}{\left(t_{k}^{}\right)}_{}^{T}]=R_{k,GAS}^{i}$. An initial state vector ${\bar{X}}_{0}^{i}=E(X_{0}^{i*})$ and an initial state vector covariance matrix ${\bar{P}}_{0}^{i}=E[X_{0}^{i*}X_{0}^{i*T}]$ are defined. Results from the method of ICEKF that combines the ISL measurement and satellite-to-GAS measurement can be obtained from:(43)$${\bar{P}}_{k}^{i}=\Phi_{k-1}^{i}{\hat{P}}_{k-1}^{i}\Phi_{k-1}^{i\;T}+G_{k-1}^{i}Q_{k-1}^{i}G_{k-1}^{i\;T}$$
(44)$$K_{k}^{i}={\bar{P}}_{k}^{i}H_{k}^{i\;T}{\left(H_{k}^{i\;T}{\bar{P}}_{k}^{i}H_{k}^{i\;}+R_{k}^{i}\right)}^{-1}$$
(45)$$\delta {\hat{x}}_{k}^{i}=K_{k}^{i}z_{k}^{i}$$
(46)$${\hat{X}}_{k}^{i}={\bar{X}}_{k}^{i}+\delta {\hat{x}}_{k}^{i}$$
(47)$${\hat{P}}_{k}^{i}=\left(I-K_{k}^{i}H_{k}^{i}\right){\bar{P}}_{k}^{i}$$

In this way, only the local state vector related to the *ith* satellite itself is included in the measurement equation. The dimension of state vector $X_{k}^{i}$ is:(48)$$N_{i}=6+D_{}^{i}$$

The computation amount for the ICEKF algorithm is:(49)$$4N_{i}^{2}(N_{i}^{2}-1)+(N_{i}^{}-1)N_{i}^{}(N_{i}^{}+1)/6+(2N_{i}^{2}+7N_{i}^{}+1)\times (m/n)$$

As a result, the computational complexity is greatly reduced. However, the state vector of the *ith* satellite is only correlated with the measurement of itself and this method must be referred to as a reduced-order suboptimal filter. In order to improve the filtering accuracy, a common approach is to iterate the process above until convergence. In a data frame, after receiving the state vectors of the other visible satellites, the *ith* satellite updates its own state vectors and covariance matrix by Equations (42)–(47). Other satellites do the same process in turn and iterate until the state vector of each satellite is converged.

However, the method of ICEKF assumes that the state vectors of the other satellites have no errors, but this is not the case. Therefore, the method of ICEKF needs an uncertain number of iterations to approach convergence. In a constellation with a large number of satellites, reaching convergence could be time-consuming [12,13].

### 5.2. Reduced-Order Increased Measurement Covariance EKF

To accelerate the data-processing in the ICEKF method, the reduced-order increased measurement covariance EKF (IMCEFK) [12] is carried out. This method includes the error of the state vectors of the *jth* satellite into the measurement covariance matrix $R_{k}^{i}$ between the *ith* satellite and the *jth* satellite. If we regenerate the ISL measurement error in Equation (42) as $\varepsilon_{k}^{i}=H_{}^{j}\delta x_{k}^{j}+\varepsilon_{}^{ij}(t_{k}^{})$, then the corresponding measurement covariance matrix is:(50)$$\begin{array}{ll} E\left[\varepsilon_{k}^{i}\varepsilon_{k}^{i}{}^{T}\right] & =E\left[(H_{k}^{j}\delta x_{k}^{j}+\varepsilon_{}^{ij}(t_{k}^{})){\left(H_{k}^{j}\delta x_{k}^{j}+\varepsilon_{}^{ij}(t_{k}^{})\right)}^{T}\right]\\ & =H_{k}^{j}{\bar{P}}_{k}^{j}H_{k}^{jT}+R_{k}^{i} \end{array}$$
where ${\bar{P}}_{k}^{j}$ is the state vector covariance matrix of the *jth* satellite. Equation (50) implies that ISL measurements contain not only the measurement errors, but also the *jth* satellite state vector error; thus, the measurement covariance matrix is assembled as:(51)$$R_{k\;assembled}^{i}=H_{k}^{j}{\bar{P}}_{k}^{j}H_{k}^{jT}+R_{k}^{i}$$
where the subscript *assembled* indicates that it is an assembled measurement covariance matrix.

Next, the $R_{k}^{i}$ in Equation (44) is replaced by $R_{k\;amp}^{i}$, and the gain matrix becomes:(52)$$K_{k}^{i}={\bar{P}}_{k}^{i}H_{k}^{iT}{\left(H_{k}^{iT}{\bar{P}}_{k}^{i}H_{k}^{i\;}+H_{k}^{j}{\bar{P}}_{k}^{j}H_{k}^{jT}+R_{k}^{i}\right)}^{-1}$$

After repeating the steps in Equations (42), (43), (51), (45)–(47), the orbits for *ith* satellites are determined. In this way, a reduction of the number of iterations is expected. The computation amount of IMCEKF is:(53)$$4N_{i}^{2}(N_{i}^{2}-1)+(N_{i}^{}-1)N_{i}^{}(N_{i}^{}+1)/6+(2N_{i}^{2}+7N_{i}^{}+1)\times (m/n)$$

In some situations, iteration may not be required [18]. To summarize, compared to ICEKF, the IMCEKF is a reduced-order approach and needs to transmit not only the local state vector but also its covariance matrix to the other satellites.

### 5.3. Balanced Extended Kalman Filter

In one computation cycle, the ICEKF and IMCEKF algorithm only improve the state vector on one end of the ISL. To increase the efficiency and accuracy, the balanced extended Kalman filter (BEKF) is proposed. For the *ith* and *jth* satellite, the satellites’ state vectors on the both ends of the ISL can be improved simultaneously. To keep the balance of the accuracy increments on both satellites, the improving state vectors should be adjusted by:(54)$${\left({\bar{P}}_{k}^{i}\right)}^{-1}\delta {\hat{x}}_{k}^{i}={\left({\bar{P}}_{k}^{j}\right)}^{-1}\delta {\hat{x}}_{k}^{j}$$

With the constraint of Equation (54), the BEKF can be derived from Equations (8), (12), and (24), and it can be completed by the following steps:(55)$${\bar{P}}_{k}^{i}=\Phi_{k-1}^{i}{\hat{P}}_{k-1}^{i}\Phi_{k-1}^{i\;T}+G_{k-1}^{i}Q_{k-1}^{i}G_{k-1}^{i\;T}$$
(56)$${\bar{P}}_{k}^{j}=\Phi_{k-1}^{j}{\hat{P}}_{k-1}^{j}\Phi_{k-1}^{j\;T}+G_{k-1}^{j}Q_{k-1}^{i}G_{k-1}^{j\;T}$$
(57)$$N_{B}^{}=\left[\begin{array}{l} H_{k}^{i}R_{}^{-1}H_{k}^{i}+{\left({\bar{P}}_{k}^{i}\right)}_{}^{-1}\;H_{k}^{i}{}^{T}R_{}^{-1}H_{k}^{j}\\ H_{k}^{j}{}^{T}R_{}^{-1}H_{k}^{i}\;H_{k}^{j}{}^{T}R_{}^{-1}H_{k}^{j}+{\left({\bar{P}}_{k}^{j}\right)}_{}^{-1} \end{array}\right]$$
(58)$$C=\left[{\left({\bar{P}}_{k}^{i}\right)}_{}^{-1}\;-{\left({\bar{P}}_{k}^{j}\right)}_{}^{-1}\right]$$
(59)$$N_{C}^{}=CN_{B}^{-1}C_{}^{T}$$
(60)$$M_{C}^{}=N_{B}^{-1}C_{}^{T}N_{C}^{-1}C$$
(61)$$K_{k}^{ij}=(I-M)\left[\begin{array}{l} {\bar{P}}_{k}^{i}H_{k}^{i\;T}\\ {\bar{P}}_{k}^{j}H_{k}^{j\;T} \end{array}\right]{\left[H_{k}^{i\;}{\bar{P}}_{k}^{i}H_{k}^{i\;T}+H_{k}^{j}{\bar{P}}_{k}^{j}H_{k}^{j\;T}+R_{k}^{i}\right]}^{-1}$$
(62)$$\left[\begin{array}{l} \delta {\hat{x}}_{k}^{i}\\ \delta {\hat{x}}_{k}^{j} \end{array}\right]=K_{k}^{}z_{k}^{i}$$
(63)$$\left[\begin{array}{l} {\hat{X}}_{k}^{i}\\ {\hat{X}}_{k}^{j} \end{array}\right]=\left[\begin{array}{l} {\bar{X}}_{k}^{i}\\ {\bar{X}}_{k}^{j} \end{array}\right]+\left[\begin{array}{l} \delta {\hat{x}}_{k}^{i}\\ \delta {\hat{x}}_{k}^{j} \end{array}\right]$$
(64)$$\left[\begin{array}{l} {\hat{P}}_{k}^{i}\;{\hat{P}}_{k}^{ij}\\ {\hat{P}}_{k}^{ji}\;{\hat{P}}_{k}^{j} \end{array}\right]=\left\{I-K_{K}^{}\left[H_{k}^{i\;}\;H_{k}^{j\;}\right]\right\}\left[\begin{array}{l} {\bar{P}}_{k}^{i}\;0\\ 0\;{\bar{P}}_{k}^{j} \end{array}\right]-M\left[\begin{array}{l} {\bar{P}}_{k}^{i}\;0\\ 0\;{\bar{P}}_{k}^{j} \end{array}\right]$$

The dimension of state vectors $X_{k}^{i}$ and $X_{k}^{j}$ of the *ith* and *jth* satellites, respectively, is:(65)$$N_{ij}^{}=12+\sum_{i=1}^{n}\left(D^{i}+D^{j}\right)$$

The computation amount of BEKF is:(66)$$4N_{i}^{2}(N_{i}^{2}-1)+(N_{i}-1)N_{i}(N_{i}+1)/6+(2N_{ij}^{2}+7N_{ij}^{}+1)\times (m/2n)$$

The method has the following features:
The method of BEKF collects the data of ISL measurements, satellite-to-GAS measurements (if they are available), and the satellite state vectors as well as their covariance matrices on both ends of the ISL. After the calculation of the BEKF algorithm, the improved state vectors and their covariance matrices are sent to the other visible satellites. The BEKF method modifies the denominator of the gain matrix $K_{k}^{ij}$ to $H_{k}^{i\;}{\bar{P}}_{k}^{i}H_{k}^{i\;T}+H_{k}^{j}{\bar{P}}_{k}^{j}H_{k}^{j\;T}+R_{k}^{i}$, which is similar to the method of IMCEKF. Furthermore, it modifies the gain matrix by a factor of $\left(I-M_{C}^{}\right)$. Therefore, the BEKF algorithm is expected to yield results with higher precision.It seems that BEKF requires more ISL processes than the other EKFs. In fact, the state vectors and their covariance matrices on both ends are improved at the same time. It is unnecessary to repeat the ISL process for the same two satellites. The computation load of BEKF is similar to that of IMCEKF.The iteration process that is implemented in the ICEFK algorithm is not required in the BEKF method to achieve high accuracy.The improving state vectors are balanced in such a way that the satellite with lower-state precision will undergo more increments in accuracy while the satellite with higher-state precision will have fewer adjustments.Compared to the other EKFs, in Equation (64), $M_{C}^{}\left[\begin{array}{l} {\bar{P}}_{k}^{i}\;0\\ 0\;{\bar{P}}_{k}^{j} \end{array}\right]$ is subtracted from the state vectors’ covariance matrices of the two satellites. Therefore, the values in the matrices are reduced and the accuracy of the state vectors is improved.The two satellites are correlated by ${\hat{P}}_{k}^{ij}$ and ${\hat{P}}_{k}^{ji}$ in Equation (65); however, these two matrices have to be ignored in this distributed filter. As a result, the current method should be categorized as a reduced-order sub-optimal orbit determination method.

## 6. Simulations and Analyses

In order to compare the performance of the abovementioned methods, navigation constellation simulations were carried out with the parameters of Walker 24/3/2:55°, 22,116 km [30]. The dynamic model applied to satellite orbit was:(1)the earth’s gravitational effects of 70 × 70,(2)the lunar, solar, and other planetary gravitational perturbations,(3)the solar radiation pressure, and(4)the other general relativistic forces.

The eighth-order Runge–Kutta method was employed for orbit integration. The IERS96 model was adopted for the Earth orientation parameters [31]. Besides, the TDMA mode was adopted in ISL with measurement frames and data transmitting frames. The total error of Ka-band ISL PR was 0.5 m (1σ). To avoid ground atmospheric disturbance, the ISLs with a vertical distance of less than 1000 km to the Earth surface were not considered. Eight GASs that are located in Xiamen, Kashi, Beijing, Lhasa, Sanya, Urumqi, Jiamusi, and Xi’an in China were set up in the simulation. With a minimum elevation of 10°, the Hopfield - Marini model [1] were employed in tropospheric delay correction for the satellite-to-GAS PR which had a total error of 3 m (1σ).

The impact of complex factors was not considered in the simulations. In addition, the ISL PR measurement noise was assumed to be normally distributed without pollution, to have better comparisons for the different algorithms.

The orbit determination simulations were carried out in two steps:(1)An analytical orbit was generated and the corresponding ISL and satellite-to-GAS PRs were calculated.(2)Using the abovementioned PRs, the satellite orbits were calculated by the different methods and compared with the analytical orbit to find out orbit determination precisions.

The position error, radial error, along track error, and cross track error versus time normalized by day for satellite SV-01 computed from different methods are presented in Figure 1, Figure 2, Figure 3 and Figure 4. It should be noted that plots of the errors from the other satellites in the constellation are excluded since the errors are similar to those of satellite SV-01.

For each method, the orbit determination errors tended to oscillate steadily after poor initial results. However, differences can be observed among the four algorithms. To have quantitative comparisons, the root mean square (RMS) of different errors for all the satellites in the constellation was calculated for the stable section. The average RMSs were then obtained by averaging the data from all the satellites and are plotted in Figure 5. It is worth pointing out that the average RMS of position error for the WCCEKF, ICEKF, IMCEKF, and BEKF algorithms was around 1.6 m, 4.5 m, 2.9 m, and 1.9 m, respectively.

Next, the computation amounts for the different methods are summarized in Table 1 and visualized in Figure 6.

The WCCEKF algorithm yielded optimal results with the highest precision. Among the three distributed orbit determination algorithms (ICEKF, IMCEKF, and BEKF), the highest orbit estimation precision was observed in the method of BEKF. When it comes to computation amounts for the four methods in Table 1 and Figure 6, the largest calculation amount was required in the WNCEKF algorithm, which achieved the best orbit accuracy. In the methods of ICEKF, IMCEKF, and BEKF with sub-optimal distributed orbit determination, significant reductions of the computation amount were observed when compared to the method of WNCEKF.

Finally, the performances of the WCCEKF, ICEKF, IMCEKF, and BEKF algorithms are summarized and provided in Table 2, considering different aspects.

## 7. Conclusions

The fundamental theory of satellite orbit determination for autonomous navigation is introduced. Four algorithms for autonomous navigation with onboard data processing, i.e., whole-constellation centralized extended Kalman filter (WCCEKF), iterative cascade extended Kalman filter (ICEKF), increased measurement covariance extended Kalman filter (IMCEKF), and balanced extended Kalman filter (BEKF), are illustrated. The WCCEFK technique processes the measurement data for orbit determination by a main satellite while the other three algorithms distribute the computation on every satellite in the constellation. The simulation results show that the method WCCEKF has the optimal orbit determination accuracy among the four methods but demands the largest computation loads. Similar computation amounts are observed in the three distributed algorithms. Compared to the ICEKF technique, covariance matrices of the other satellites are absorbed into the measurement covariance matrix in the method of IMCEKF. As a result, a smaller orbit error is observed in the IMCEKF algorithm than in the ICEKF. Since the BEKF method estimates the state vectors of the satellites on both ends of the ISL in a balanced mean and increases their accuracy simultaneously, this method yields the best results among the three distributed estimation algorithms. The BEKF can be considered as an appropriate distributed data processing algorithm for a GNSS.

## Figures and Tables

**Figure 1 sensors-19-01031-f001:**
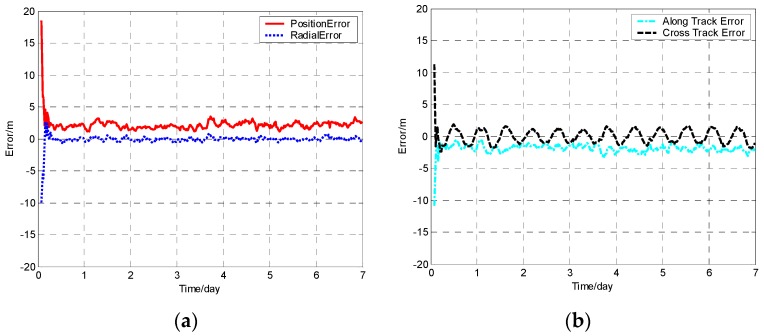
Orbit determination errors of SV-01 by the whole-constellation centralized extended Kalman filter (WCCEKF) algorithm. (**a**) the position error and radial error; (**b**) along track error and cross track error.

**Figure 2 sensors-19-01031-f002:**
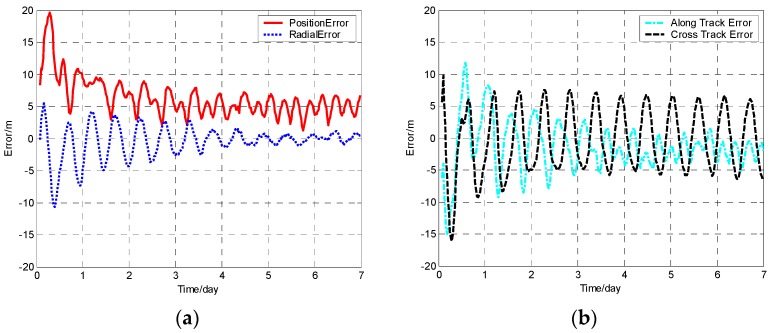
Orbit determination errors of SV-01 by the iterative cascade extended Kalman filter (ICEKF) algorithm. (**a**) the position error and radial error; (**b**) along track error and cross track error.

**Figure 3 sensors-19-01031-f003:**
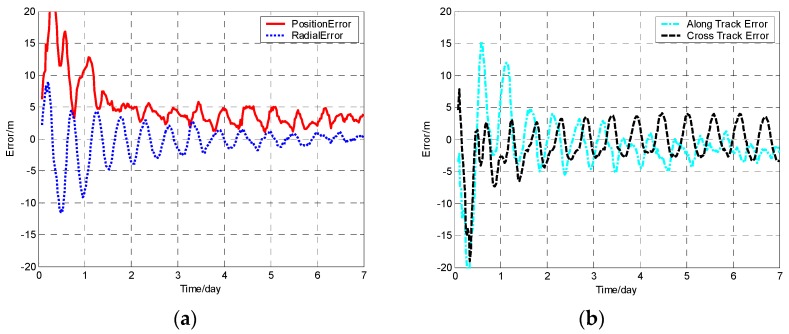
Orbit determination errors of SV-01 by the increased measurement covariance extended Kalman filter (IMCEKF) algorithm. (**a**) the position error and radial error; (**b**) along track error and cross track error.

**Figure 4 sensors-19-01031-f004:**
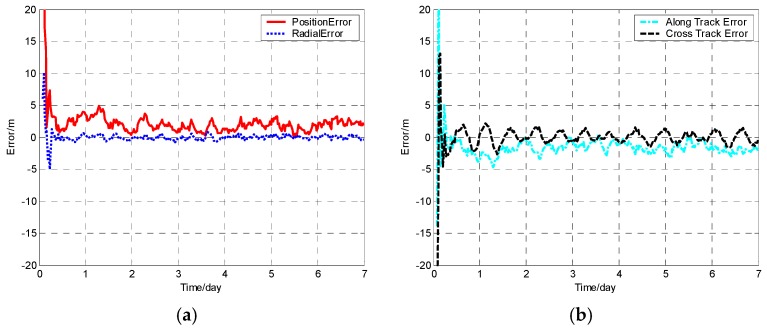
Orbit determination errors of SV-01 by the balanced extended Kalman filter (BEKF) algorithm. (**a**) the position error and radial error; (**b**) along track error and cross track error.

**Figure 5 sensors-19-01031-f005:**
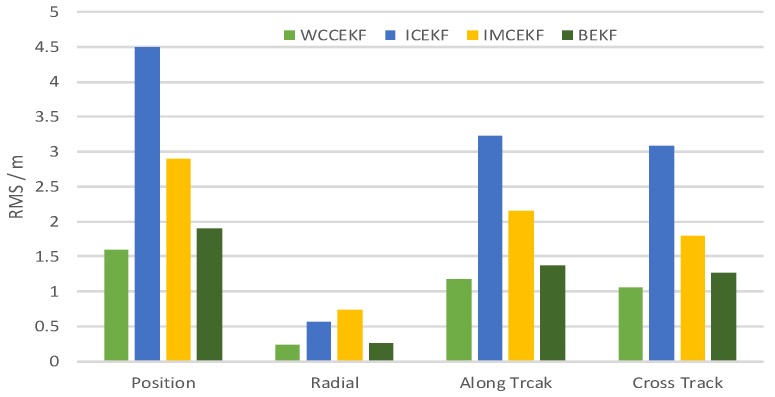
Average RMS of orbit errors.

**Figure 6 sensors-19-01031-f006:**
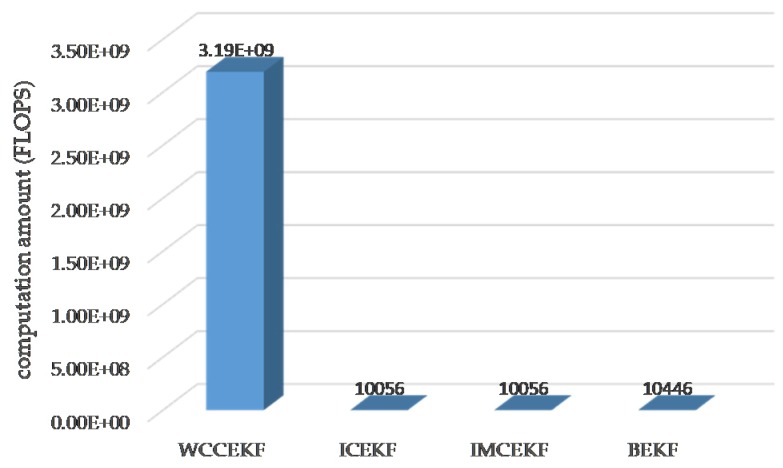
Computation amounts.

**Table 1 sensors-19-01031-t001:** Computation amounts, unit: FLOPS (floating-point operations per second).

Methods	Computation Amount (FLOPS: Floating-Point Operations Per Second)
WCCEKF	$4N_{W}^{2}(N_{W}^{2}-1)+(N_{W}^{}-1)N_{W}^{}(N_{W}^{}+1)/6$ + $(2N_{W}^{2}+7N_{W}^{}+1)\times m$
ICEKF or IMCEKF	$4N_{i}^{2}(N_{i}^{2}-1)+(N_{i}^{}-1)N_{i}^{}(N_{i}^{}+1)/6$ + $\left(2N_{i}^{2}+7N_{i}^{}+1)\times (m/n\right)$
BEKF	$4N_{i}^{2}(N_{i}^{2}-1)+(N_{i}^{}-1)N_{i}^{}(N_{i}^{}+1)/6$ + $\left(2N_{ij}^{2}+7N_{ij}^{}+1)\times (m/2n\right)$

**Table 2 sensors-19-01031-t002:** Comparisons of performances.

Algorithm	Description	Computation Amount	Orbital Accuracy
WCCEKF	Whole-constellation centralized EKF	Maximum	Best
ICEKF	Iterative cascade EKF	Minimum	Normal
IMCEKF	Increased measurement covariance EKF	Minimum	Normal
BEKF	Balanced EKF	Minimum	Better

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
