# Peer review of "Distributed Orbit Determination for Global Navigation Satellite System with Inter-Satellite Link"

_sensors, 2019, doi:10.3390/s19051031_

Reviewer 1 Report

The paper present a new algorithm for the determination of GNSS orbit by means of a distribuited algorithm. 

The proposed algorithm seems to perform better than other distribuited algorithms and with comparable computation burden.

The paper is clear and well written, only some minor typos are present (see for instance pag. 14 lines 414 - 415 "algorithms in are summarized"), which however in my opinion can be corrected within the proofread step.

Author Response

Dear reviewer

Thank you for the reviewing and pointing out the flaws in the manuscript.

We have located a few typos in the manuscript. The word "in" in lines 414 – 415 on page14 has been removed.

Yours Sincerely,
Yuanlan Wen
On behalf of theco-authors

Reviewer 2 Report

Dear authors, I've read the paper with interest and is well structured and written. 

Just a comment is related to the results presentation

I agree with you about the decision to show just an example in graphical way because you say that the obtained results for all satellites orbit determination are almost comparable, but when you summarize the results using statistics parameters should be better to present RMS considering all satellites considered for the simulation. In such way could be much more clear that the statistics are not related to the best (or the worst) satellite.

Pleas consider to change this section. 

Author Response

Dear reviewer,

Thank you for the reviewing and pointing out the flaws in the manuscript.

The RMSs presented in the manuscript are actually statistical results by averaging the data from all the satellintes in simluation consetallation. However, the idea was not able to be stated very clearly in the original manuscript. Revisions have been made as follows:

page. 12 lines 386 – 387: It should be noted that plots of the errors from the other satellites in the constellation are excluded since the errors are similar to that of satellite SV-01.

page. 13 lines 392 – 395: To have quantitative comparisons, the root mean square (RMS) of different errors for all the satellites in the constellation is calculated in stable section. The average RMSs are then obtained by averaging the data from all the satellites and plotted in Figure 5.

The modifications were highlighted in red in the revised manuscript.

Sincerely yours,
Yuanlan Wen
On behalf of the co-authors
